# Patient attitudes towards faecal sampling for gut microbiome studies and clinical care reveal positive engagement and room for improvement

**Laura A. Bolte** [1,2‡], **Marjolein A. Y. Klaassen** [1,2‡], **Valerie Collij** [1,2], **Arnau Vich Vila** [1,2], **Jingyuan Fu** [1,3], **Taco A. van der Meulen** [4], **Jacco J. de Haan** [5], **Gerbrig J. Versteegen** [6], **Aafje Dotinga** [7], **Alexandra Zhernakova** [2], **Cisca Wijmenga** [2], **Rinse K. Weersma** [1], **Floris Imhann** [1,2]*

1 Department of Gastroenterology and Hepatology, University of Groningen, University Medical Center Groningen, Groningen, The Netherlands, 2 Department of Genetics, University of Groningen, University Medical Center Groningen, Groningen, The Netherlands, 3 Department of Pediatrics, University of Groningen, University Medical Center Groningen, Groningen, The Netherlands, 4 Department of Oral and Maxillofacial Surgery, University of Groningen, University Medical Center Groningen, Groningen, The Netherlands, 5 Department of Medical Oncology, University of Groningen, University Medical Center Groningen, Groningen, The Netherlands, 6 Department of Medical Psychology, University of Groningen, University Medical Center Groningen, Groningen, The Netherlands, 7 Lifelines Cohort Study, Groningen, The Netherlands

‡ These authors share first authorship on this work.
* f.imhann@rug.nl

**Data Availability Statement:** The summary statistics for each question in the questionnaire are available in Supplementary Tables S1 and S2. Using these statistics, all tests in this study can be

## Abstract

Faecal sample collection is crucial for gut microbiome research and its clinical applications. However, while patients and healthy volunteers are routinely asked to provide stool samples, their attitudes towards sampling remain largely unknown. Here, we investigate the attitudes of 780 Dutch patients, including participants in a large Inflammatory Bowel Disease (IBD) gut microbiome cohort and population controls, in order to identify barriers to sample collection and provide recommendations for gut microbiome researchers and clinicians. We sent questionnaires to 660 IBD patients and 112 patients with other disorders who had previously been approached to participate in gut microbiome studies. We also conducted 478 brief interviews with participants in our general population cohort who had collected stool samples. Statistical analysis of the data was performed using R. 97.4% of respondents reported that they had willingly participated in stool sample collection for gut microbiome research, and most respondents (82.9%) and interviewees (95.6%) indicated willingness to participate again, with their motivations for participating being mainly altruistic (57.0%). Responses indicated that storing stool samples in the home freezer for a prolonged time was the main barrier to participation (52.6%), but clear explanations of the sampling procedures and their purpose increased participant willingness to collect and freeze samples (P = 0.046, P = 0.003). To account for participant concerns, gut microbiome researchers establishing cohorts and clinicians trying new faecal tests should provide clear instructions, explain the rationale behind their protocol, consider providing a small freezer and inform patients about study outcomes. By assessing the attitudes, motives and barriers

replicated. Our cohorts comprise specific groups of patients from a specific geographical area and the content of the questionnaire comprises household composition, education level and home details about the toilet and the freezer. As a consequence, individual level data could lead to identification of patients. Therefore, European privacy law prohibits us from making individual level data publicly available. Data requests should be sent to the Research Office Lifelines (research@lifelines.nl) and the Medical Ethics Review Board of the UMCG (metc@umcg.nl).

**Funding:** LAB and RKW are supported by a research grant from the Seerave Foundation. RKW is supported by the collaborative TIMID project (LSHM18057-SGF) financed by the PPP allowance made available by Top Sector Life Sciences & Health to Samenwerkende Gezondheidsfondsen (SGF) to stimulate public-private partnerships and co-financing by health foundations that are part of the SGF. JF and AZ are supported by the Gravitation grant ExposomeNL from the from the Dutch Organization for Scientific Research (Nederlandse Organisatie voor Wetenschappelijk Onderzoek, NWO) (024.004.017). AZ is further supported by a European Research Council (ERC) Starting Grant (715772) and a NWO-VIDI grant (016.178.056). JF is supported by the NWO Gravitation Netherlands Organ-on-Chip Initiative (024.003.001) and the ERC Consolidator grant (101001678). JF and AZ are further supported by a grant from the Dutch Heart Foundation (CardioVasculair Onderzoek Nederland, CVON) (2018-27). CW is supported by the NWO Gravitation grant (024.003.001), a Spinoza award (NWO SPI 92-266) and a grant from the Netherlands' Top Institute Food and Nutrition (GH001).

**Competing interests:** Floris Imhann received a speaker fee from Abbvie. Rinse Weersma received speaker fees from Abbvie, MSD, and Boston Scientific, a consulting fee from Takeda Pharmaceuticals and unrestricted research grants from Pfizer, Takeda, Ferring and Tramedico. This does not alter our adherence to PLOS ONE policies on sharing data and materials.

surrounding participation in faecal sample collection, we provide important information that will contribute to the success of gut microbiome research and its near-future clinical applications.

## Introduction

Gut microbiome research is being conducted using ever greater sample sizes to elucidate the role of gut microbiota in the pathogenesis of Inflammatory Bowel Diseases (IBD) and other immune-mediated inflammatory diseases [1–4]. The results of these studies hold great promise for clinical applications that use microbiome features as diagnostic biomarkers [5], determinants of disease activity [3] and predictors of individual drug response [6, 7]. The microbiome itself may also be a treatment target for prebiotic, probiotic, antibiotic and dietary interventions [2, 8, 9]. Moreover, as clinical interest grows in the use of faecal microbiota transplantation (FMT) to treat dysbiosis-related disorders such as recurrent *Clostridium difficile*–associated diarrhoea and IBD, so will the need for voluntary stool donors [10–12].

As a consequence, there is a growing demand for stool samples collected by both patients and healthy volunteers. However, little is known about participant perspectives on collecting faecal samples for microbiome research and future care. Several studies have examined participant experiences with the faecal occult blood test (FOBT) used in colorectal cancer screening, the results of which mainly capture experiences coloured by the fear of having cancer [13, 14]. Other studies report barriers to faecal sample collection in general practice, including difficulty with the process, embarrassment and concerns around hygiene [14, 15]. Despite these barriers, most patients in the clinical setting do provide faecal samples because they are unwell and it has been recommended that they do so [15]. While personal benefit has been identified as the main motive for collecting and returning a stool sample in clinical care [14], there is no direct personal benefit for voluntary stool donors for FMT or microbiome research, who may face similar barriers.

In contrast to FMT or clinical tests such as the FOBT, at-home collection of faecal samples for microbiome research requires participants to follow sampling protocols and to store the sample in their home freezer in order to avoid post-collection bias in microbial composition, and this storage aspect may present an additional hurdle for volunteers. The accepted best-practices for microbiome studies involve freezing the sample to -80˚C within 15 minutes of production and storage in a domestic frost-free freezer for fewer than 3 days. Samples taken for metabolomics studies, in particular, require that stool be frozen without preservatives and the freezing of live bacteria in glycerol preservative for culturing [16–18]. Since the stool samples used for research are collected by IBD patients at home, researchers need these patients to fully understand how to collect the sample. However, patient willingness to provide a faecal sample or to store it in the home freezer for research, their motives for and experiences with participation in microbiome research and the potential barriers they encounter, or how these barriers can be overcome, have thus far not been described.

Here, we explore the motives for and barriers to faecal sample collection given by 780 patients and healthy volunteers, including participants of one of the largest IBD gut microbiome cohorts to date. Our findings allow us to make recommendations for researchers and clinicians that will allow them to better account for participant attitudes when designing gut microbiome studies for research and clinical applications.

## Methods

### Cohorts and participants

In total, we contacted 1250 individuals, including IBD patients, patients with other disorders and healthy volunteers. A questionnaire (S1 Table) was sent in January 2017 to 772 patients who had previously been recruited at the University Medical Center Groningen in the Netherlands for gut microbiome studies for which they needed to provide a faecal sample. These patients had been included in four disease-specific cohorts for IBD (n = 660), melanoma (n = 9), Sjögren's syndrome (n = 55) and systemic lupus erythematosus (SLE) (n = 48) (Fig 1). The latter three cohorts only comprised the participants who joined the gut microbiome studies. The questionnaire was aimed at obtaining patient experiences and identifying barriers encountered during the collection process. With the IBD cohort, we were also able to send out questionnaires to patients who previously refused to participate in gut microbiome research. The questionnaire recipients in the IBD cohort therefore comprised both patients who were previously willing to collect a stool sample for research (n = 577, IBD-Willing) and those who were not willing to do so (n = 83, IBD-Unwilling), indicating a willingness rate of 87.4% of the IBD microbiome study prior to this survey.

In addition, we interviewed a random selection of participants (n = 478) from the Lifelines general population cohort, of whom 9,547 individuals participated in the faecal sample collection project DAG3, using a brief questionnaire to analyse their opinions in the faecal sampling collection process (S2 Table) [19].

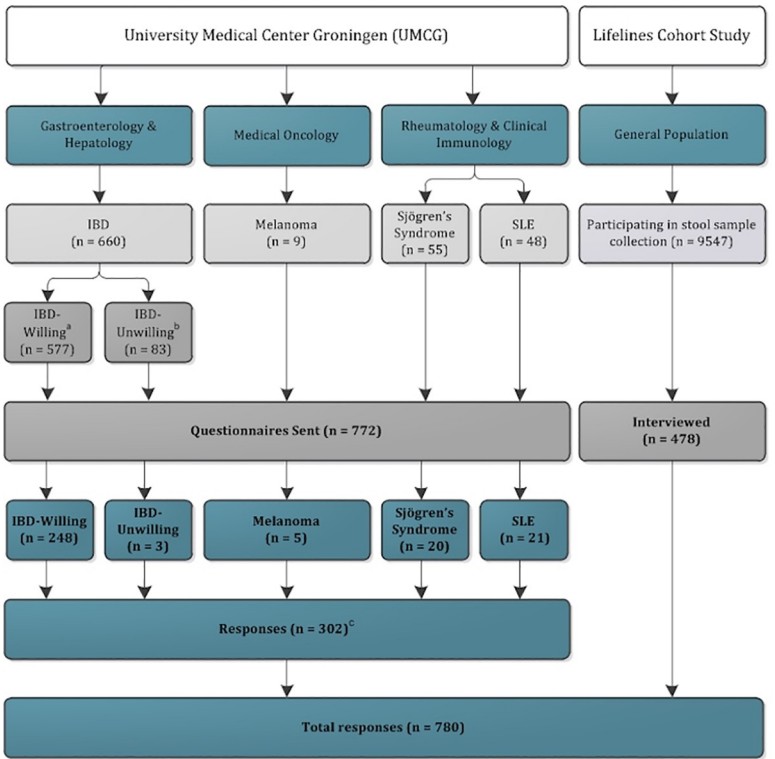

**Fig 1. Cohort selection and responses.** Chart depicts the cohorts, diseases, departments and respondents in this study. IBD inflammatory bowel disease; SLE systemic lupus erythematosus; n number. From top to bottom: Source, Department, Cohort, Sub-cohort, Assessment Method, Responses by Cohort, Responses to Questionnaire, Total Responses. [a] IBD-Willing: patients who previously indicated their willingness to collect faecal samples for research. [b] IBD-Unwilling: patients who previously indicated that they were not willing to collect faecal samples for research. [c] Total responses include 5 individuals who did not fill in their participation number and could not be assigned to a cohort.

## Ethics approval and consent to participate

The collection of faecal samples was previously approved by the Institutional Review Board (IRB) of the University Medical Center Groningen (IRB number 2008.338). All participants who participated in the faecal sample collection studies provided a signed informed consent form. For a single questionnaire study, no additional IRB approval was required according to Dutch medical research law. The questionnaire and the interview to assess the attitudes towards faecal sampling of patients and the general population, respectively, have been designed specifically for this study and are not published elsewhere. Consent to participate was integrated in the questionnaire.

## Questionnaire design and processing

The questionnaire was designed in collaboration with a psychologist from the IBD Centre in Groningen. It covered eight distinct areas: (A) general information, including living situation, (B) prior experiences with faecal sample collection, (C) information about the type of toilet and freezer at home, (D) perceptions of the collection process, (E) perceptions of storing faecal samples in their freezer, (F) experience with the pick-up of the faecal samples from the participant's home by hospital employees, (G) satisfaction with the information provided by our university medical centre and (H) future willingness to collect faecal samples for clinical care purposes. An English translation of the Dutch questions and the answers to the questionnaire and the interview can be found in the S1 and S2 Tables, respectively.

In our questionnaire, we addressed both patients previously willing to participate in faecal sample collection for microbiome research (IBD-Willing, melanoma, SLE and Sjögren's Syndrome) and patients not willing to participate (IBD-Unwilling). The IBD-Unwilling cohort was asked to answer questions about their reasons for not participating despite their willingness to participate in research in general. Patients who had participated in faecal sample collection for research were asked about their experiences. Of the 347 respondents to our questionnaire, 45 gave inconsistent answers to questions, indicating they had not correctly understood the instructions. We chose to exclude these 45 questionnaires, resulting in a final sample of 302 respondents (39.1% response rate). To ensure that exclusion of these 45 questionnaires did not introduce bias, we performed our analyses on both the full set (347) and the final set (302) for comparison purposes and found similar results.

## Statistical analyses

Descriptive statistics were determined for each question using the statistical software package R [20] (S1 Table). Chi-Square tests and Fisher's exact tests were performed to determine statistically significant differences between counts (Table 1).

The following five associations were calculated:

1. Willingness to collect faecal samples for future screening and care vs. Gastrointestinal disease (Fisher's exact test) to test if disease presence (Gastrointestinal disease or No gastrointestinal disease) is associated with willingness;

2. Willingness to collect faecal samples for future screening and care vs. Home situation (Fisher's exact test) to test if having co-habitants is associated with willingness;

3. Willingness to collect faecal samples for future screening and care vs. Clarity of the instruction manual (Fisher's exact test) to test if understanding the protocol properly is associated with willingness;

**Table 1. Patient willingness to collect and freeze faecal samples and associated factors.**

| | | n (%) | | |
|---|---|---|---|---|
| **Motivation to participate in faecal sample collection for microbiome research** | | **All patients** | | |
| | Benefit for other patients | 170 (57.0%) | | |
| | Both, benefit for self and others | 48 (16.1%) | | |
| | Benefit for self | 38 (12.8%) | | |
| | Other options/ combinations | 27 (9.1%) | | |
| | Did not fill in | 15 (5.0%) | | |
| | Total | 298 (100%) | | |
| **Willing to collect faecal samples for future healthcare** | **Willing to collect** | **IBD cohort** | **Non-IBD cohort** | |
| **Split by IBD / No IBD** | Yes | 224 (89.6%) | 43 (89.6%) | P = 0.673 |
| | No | 15 (6.0%) | 4 (8.3%) | |
| | Did not fill in | 11 (4.4%) | 1 (2.1%) | |
| | Total | 250 (100%) | 48 (100%) | |
| **Willing to collect faecal samples for future healthcare** | **Willing to collect** | **GI-disorder** | **No GI-disorder** | |
| **Split by GI-disorder / No GI-disorder** | | | | |
| | Yes | 205 (81.0%) | 38 (90.5%) | |
| | No | 34 (13.4%) | 4 (9.5%) | P = 0.564 |
| | Did not fill in | 14 (5.5%) | 0 (0.0%) | |
| | Total | 253 (100%) | 42 (100%) | |
| **Willing to collect for future healthcare** | **Willing to collect** | **Living alone** | **Living together** | |
| **Split by living alone/living together** | Yes | 49 (16.4%) | 213 (71.5%) | P = 0.543 |
| | No | 2 (0.7%) | 16 (5.4%) | |
| | Did not fill in | | 18 (6.0%) | |
| | Total | | 298 (100%) | |
| **Was the collection process easy?** | | **All patients** | | |
| | Yes | 253 (84.9%) | | |
| | No | 35 (11.7%) | | |
| | Did not fill in | 10 (3.4%) | | |
| | Total | 298 (100%) | | |
| **Time between sample collection and storage in the freezer** | | **All patients** | | |
| | 1–5 minutes | 186 (62.4%) | | |
| | 5–10 minutes | 74 (24.8%) | | |
| | 10–15 minutes | 20 (6.7%) | | |
| | >15 minutes | 4 (1.3%) | | |
| | Did not fill in | 14 (4.7%) | | |
| | Total | 298 (100%) | | |
| **Unpleasant to store faecal samples in home freezer?** | | **All patients** | | |
| | Yes | 73 (24.5%) | | |
| | No | 215 (72.1%) | | |
| | No answer | 10 (3.4%) | | |
| | Total | 298 (100%) | | |

(*Continued*)

**Table 1.** (Continued)

| | | n (%) | | |
|---|---|---|---|---|
| **Maximum time patients want to store faecal samples in their freezer** | | **All patients** | | |
| | I do not want that | 2 (0.9%) | | |
| | 1 to 3 days | 33 (14.6%) | | |
| | 1 week | 57 (25.3%) | | |
| | 2 to 4 weeks | 25 (11.1%) | | |
| | >1 month | 6 (2.7%) | | |
| | I do not mind | 92 (40.9%) | | |
| | No answer | 10 (4.4%) | | |
| | Total | 225 (100%) | | |
| **Was it clear why faecal samples need to be frozen?** | | **All patients** | | |
| | Yes | 224 (75.2%) | | |
| | No | 57 (19.1%) | | |
| | Did not fill in | 17 (5.7%) | | |
| | Total | 298 (100%) | | |
| **Clarity of instruction manual vs. Willing to collect faecal samples** | **Clarity of instruction** | **Willing to collect** | **Not willing to collect** | |
| | Yes, very clear | 95 (31.9%) | 5 (1.7%) | |
| | Yes, clear | 157 (52.7%) | 11 (3.7%) | |
| | Neither clear nor unclear | 8 (2.7%) | 1 (0.3%) | |
| | No, unclear | 4 (1.3%) | 1 (0.3%) | P = 0.046 |
| | No, very unclear | 0 (0.0%) | 1 (0.3%) | |
| | Did not fill in | 15 (5.0%) | | |
| | Total | 298 (100%) | | |
| **Knowing the purpose of freezing vs. Willing to freeze** | **Willing to freeze** | **Knowing the purpose of freezing** | **Not knowing the purpose of freezing** | |
| | Willing to freeze | 200 (67.1%) | 42 (14.1%) | P = 0.003 |
| | Not willing to freeze | 23 (7.7%) | 15 (5.0%) | |
| | Did not fill in | 18 (6.0%) | | |
| | Total | 298 (100%) | | |

IBD inflammatory bowel disease; GI gastrointestinal; SLE systemic lupus erythematosus; n number; % percentage of total

4. Willingness to collect faecal samples for future screening and care vs. Clarity of oral instruction (Fisher's exact test) to test if understanding the protocol properly is associated with willingness;

5. Willingness to store faecal samples in the home freezer for future screening and care vs. Knowing the purpose of freezing the samples (Chi-Square test of independence with Yate's continuity correction) to test if understanding the reason for freezing is associated to increased willingness of storing the samples in the home freezer.

## Results

Of the 772 patients who received the questionnaire, 302 patients responded (39.1%). When combined with the 478 Lifelines interviewees, we had information from 780 individuals in total (Fig 1).

Of the patients who responded to the written questionnaire, 97.4% had collected a faecal sample for prior gut microbiome research projects. Unfortunately, response from

the IBD patients who did not want to participate in gut microbiome research was very low: only three of the 83 IBD-Unwilling patients responded to the questionnaire, making it hard to draw broad conclusions from their answers. Nevertheless, extensive and valuable information could be obtained from the participants who did respond (Table 1, S1 Table).

Respondent motivations for participating in research projects were mainly altruistic, as future benefits for other patients (57.0%) was mentioned much more often than future benefits for themselves (12.8%) or future benefits for both themselves and others (16.1%). Most of the patients who responded (82.9%) and the population controls who were interviewed (95.6%) indicated that they were willing to collect faecal samples for future screening or research. We had anticipated that respondents with gastrointestinal disorders, who are more accustomed to handling stool, would be more willing to collect a stool sample. However, we found that having a gastrointestinal disorder was not related to the willingness to do so, with all groups showing similarly high levels of willingness to participate in future collections (IBD-cohort, willing: 224 of 250 (89.6%) vs. Non-IBD-cohort, willing: 43 of 48 (89.6%), P = 0.673, Fisher's exact test). Only 26.2% of the patients who responded felt the collection of faecal samples was dirty, and most of the population controls interviewed perceived faecal sample collection as 'not inconvenient at all' (49.8%) or 'not inconvenient' (28.7%).

Most patients thought the collection process was easy (84.9%), immediately succeeded in collecting the sample (89.0%) and were able to store their faecal sample in the freezer within 15 minutes (93.9%) as required, with 62.4% of these respondents reporting only needing 5 minutes to do so. Most respondents (72.1%) did not mind storing the stool samples in their home freezer. However, while most patients were willing to store a stool sample in their freezer, many were only willing to do so for a brief period of time: maximum 1 to 3 days (14.6%), 1 week (25.3%), or 2 to 4 weeks (11.1%). 40.9% said that they did not mind storing faecal samples for a longer time. Some patients even reported clearing the entire freezer before the stool sample collection and keeping it empty until the sample was picked up on dry ice by our collection team.

Household composition did not influence willingness to collect and store stool samples in a home freezer, as we saw no difference in attitude between participants living alone versus those living with a partner, children, parents or roommates (P = 0.543, Fisher's exact test). A minority of respondents (19.1%) did not understand why the faecal sample needed to be frozen. This is an important observation because the clarity of the written instructions was associated with future willingness to collect stool samples (P = 0.046, Fisher's test) and knowing the purpose of freezing stool (stopping bacterial growth) was associated with future willingness to freeze the stool samples (P = 0.003, Chi-square test).

More than half of the patients (58.3%) did not know how the stool samples would be processed and investigated, even though most patients (80.2%) indicated that they would like to learn more about the results of the gut microbiome research they were participating in, and some felt very disappointed about not being briefed afterwards.

## Discussion

In this study, we investigated the attitudes towards faecal sampling of participants in one of the largest IBD gut microbiome cohorts and compared them to those of other patient cohorts and healthy volunteers [21]. By assessing the attitudes, motives and barriers surrounding participation in faecal sample collection, we are able to provide important information that will contribute to the success of gut microbiome research and its near-future clinical applications.

### Gut microbiome researchers setting up new cohorts or clinicians trying new faecal tests should not shy away from doing so and should focus on providing adequate subject information

Our study demonstrates that stool sample collection for gut microbiome studies and future clinical applications is acceptable to the majority of IBD patients and even to population controls. Most IBD patients (87.4%) were willing to participate in our previous stool sample collection (IBD-Willing, n = 577), and most respondents (82.9%) and interviewees (95.6%) indicated that they were willing to participate again.

Other studies have assessed patient willingness to receive or donate stool samples for FMT, the transfer of faecal material containing microbiota from a healthy donor into a diseased patient. One study found that 77% of patients visiting the gastroenterologist would undergo FMT if medically indicated [22], whereas only 36.9% of IBD patients were willing to undergo FMT in a report by Zeitz et al. [23]. Familiarity with the gut microbiome might contribute to the higher willingness to participate in gut microbiome research in our study compared to FMT. Previous studies had found that only 46.5% of IBD patients [23] and 12% of patients visiting a gastroenterologist [22] knew about FMT. Interestingly, the willingness rate of the IBD patients to undergo FMT almost doubled after an information leaflet was provided [23]. Recognition of FMT in postgraduate medical students has been shown to be similarly low [24]. While nearly half of these students had not heard about FMT, the majority recognised that disrupting and restoring the gut microbiota played an important role in the pathogenesis and prevention of diseases. In the same study, willingness to undergo FMT or donate samples was significantly higher among those who were familiar with FMT [24].

### Researchers and clinicians should inform participating patients and healthy volunteers about the outcome of the research

In our study, patients were very interested in the outcome of the study they contributed to and were disappointed when they were not informed about the results. Most of our participants also indicated a desire to know more about the study and its outcome. This is in line with a previous report of the attitudes of 400 patients towards participation in clinical trials conducted at an internal medicine ward [25]. Positive feedback on how FMT can help patients has also been shown to be a motivator for donating faecal samples for FMT [10]. Based on the responses to our questionnaire, our team of microbiome researchers wrote a newsletter for participants about our scientific findings and publications. We recommend future researchers and clinicians provide similar feedback when possible.

### An emphasis on the public benefit of the research could help with establishing large cohorts for microbiome research

The main driver for participation in gut microbiome research reported by our respondents was the possibility that the research could benefit others with disease (57%). The motivation to contribute to research for the next generation of patients affected by the disease has also been reported to rank highly in other studies of research participation [26, 27]. McSweeney et al. also identified altruism as the main motive to donate faecal samples for FMT, and many patients who were willing to donate faecal samples said they did so to help those who were ill and to contribute to progress in scientific research [10]. As expected, this differs from the motives of patients who collect or receive faecal samples as part of their clinical care, where personal benefit is the main incentive [14]. Despite concerns around hygiene, logistics and privacy, most patients return their faecal sample to their doctor because it was recommended that they do so for their own health [15]. Similarly, other studies have shown that the majority of patients would undergo FMT if it

was medically indicated and recommended by the doctor [22] and that willingness to undergo FMT was positively associated with disease severity and previous TNF-treatment in IBD patients [23, 28]. These factors are not relevant when recruiting volunteers for microbiome research or healthy donors for FMT, making it even more crucial to inform potential volunteers about the process and to remove any barriers in order to obtain sufficient sample sizes or guarantee cost-effectiveness. Our study suggests that an emphasis on the public benefit of the research could help with establishing large cohorts for microbiome research [27].

### In studies where a time-series of many stool samples needs to be collected, researchers should consider providing participants with a small freezer

Only a minority of our participants (26.2%) felt the collection of faecal samples was dirty or inconvenient. However, the need to store samples in a participant's home freezer can be a barrier to participation in faecal sample collection, especially when participants have to store samples for a prolonged period. While most patients were willing to store a stool sample in their freezer (72.1%), many were only willing to do so for a brief period of time, from a few days to a maximum of 1 month. In another study in which patients were interviewed about providing faecal samples to their general practitioner, a much larger proportion of patients mentioned embarrassment and concerns about hygiene and contamination, discretion and privacy [14]. Fear of infectious diseases and disgust about the procedure were also identified as the most common concerns of patients about undergoing FMT [22, 23]. This is underlined by the finding that IBD patients would choose a colonoscopy as the preferred route of FMT rather than an enema or nasogastric tube [22, 23]. Even post-graduate medical students considered donating faeces troublesome because it hampered their privacy, and they also expressed concerns about the acceptability among patients [24]. Privacy was not a big concern in our study. While screening for FMT requires the donor to provide a lot of private information, as not all stool samples are suitable, participants in gut microbiome research might feel more anonymous. We also hypothesized that IBD patients are more accustomed to handling faecal samples, resulting in fewer perceived barriers. However, we could not find significant differences in the willingness to collect and store faecal samples between our IBD cohort and the other cohorts.

### Collection process perceived as easy

Most patients thought the collection process was easy (84.9%) and reported that they immediately succeeded in collecting the sample (89.0%) and in storing it in the freezer within 15 minutes according to the collection protocol (93.9%), which indicates that faecal sampling does not present a significant logistical challenge for individuals. Other studies have identified barriers towards faecal sampling in clinical care, including difficulty with the collection process, lack of information given by doctors and inability to return the sample to the institution [14, 15]. The difficulty of collecting the faecal sample was one of the major factors impacting FOBT response in a South African study [29]. Higher donation frequency, the logistics of collection or transport of faecal samples, the screening process, lack of public awareness and negative social perception have also been identified as deterrents to donating stool for FMT [10]. One reason why most of our participants perceived the collection system as easy could be that we provided an instruction sheet with the collection kit.

### Gut microbiome researchers and clinicians should explain why their collection protocol was designed in a specific way

We show that understanding the purpose of our procedures is associated with increased willingness to collect and freeze stool samples. Explaining the procedures and the reasons why

they need to be carried out in a specific way increases participant willingness to collect and freeze a faecal sample. This may be particularly important in populations with lower health literacy, with one study showing that higher education levels are strong predictors of FMT acceptance in patients [22]. Similar to our findings, other studies have indicated that patients collecting faecal samples for clinical care value an information leaflet provided with the stool collection kit [14] and that screening compliance for FOBT is significantly improved when patients obtain this information [30].

## Household composition did not affect willingness to collect and store a faecal sample

We saw no significant difference in attitude between participants living alone versus those living with partners, children, parents or roommates. Of those willing to collect faecal samples for research, 16.4% lived alone and 71.5% lived with others. Another study even showed that having children and being married were strong predictors of FMT acceptance in patients [22]. It is possible that patients with children are more likely to embrace FMT, even though it may be unappealing in nature, because of their responsibility towards their family [27].

## Strengths and limitations

Our questionnaire study was limited by knowing only the answers of the respondents. IBD patients who previously declined to participate in our gut microbiome studies (IBD-Unwilling, n = 83) were also less likely to respond to the questionnaire (n = 3), making it difficult to assess their reasons for refusing participation. Overall, the 39.1% response rate to our questionnaire is in line with the recognised 40% average response rate for postal surveys [31]. Another survey of IBD patients investigating their perspectives on FMT obtained a similar response rate of 31.4% [23]. The positive attitudes towards faecal sample collection in our study may not always be representative of other patients, and attitudes may differ depending on the reason for stool sample collection, e.g. samples collected for research vs. those collected for diagnosis of a potential disease (a process that may be accompanied by fear), or the health care setting, e.g. secondary vs. routine primary care. The strength of our study is that we were able to obtain information on the attitudes, motives and barriers surrounding participation in gut microbiome research for 780 patients with different disorders and for healthy volunteers, a group who have not been assessed to date. We obtained enough information to formulate the following conclusions and recommendations for both gut microbiome researchers and clinicians.

## Conclusions

Targeting the gut microbiome will soon be part of the diagnostic process and treatment of IBD and other diseases associated with microbial dysbiosis [5, 6, 32, 33], requiring repeated sampling from patients [34]. Here, we assessed the perspectives of patients and healthy volunteers on faecal sampling for gut microbiome research.

We derive the following recommendations for gut microbiome researchers and clinicians:

1. Gut microbiome researchers setting up new cohorts and clinicians trying new faecal tests should not shy away from doing so.

2. Gut microbiome researchers and clinicians should explain to participants why their collection protocol was designed in a specific way.

3. In studies where a time-series of many stool samples needs to be collected, researchers should consider providing participants with a small freezer.

4. Researchers and clinicians should inform participating patients and healthy volunteers about the outcome of the research.

## Supporting information

**S1 Table. Patient questionnaire outcome.** Translation of the questionnaire and descriptive statistics for each question. IBD inflammatory bowel disease; SLE systemic lupus erythematosus; n number; % percentage of total. a. IBD-Willing: IBD patients who previously indicated willingness to collect faecal samples for research. b. IBD-Unwilling: IBD patients who previously indicated not being willing to collect faecal samples for research. c. No ID-number: patients who did not fill in their participation number and could not be assigned to a cohort. d. All-Willing: all patients who previously indicated willingness to collect faecal samples for research, i.e. all except for the IBD-Unwilling cohort. e. All-Unwilling: all patients who previously indicated not being willing to collect faecal samples for research.
(XLSX)

**S2 Table. Interview outcome.** Translation of the interview and descriptive statistics for each question.
(XLSX)

**S1 File. Supplementary methods.** Additional information on statistical analyses and results of the interview conducted in the general population cohort.
(DOCX)

## Acknowledgments

We would like to thank the participants of this study for filling in the questionnaire and participating in the interviews, Lifelines for providing data on the population control participants, Wilma Westerhuis for collection logistics, Esther Bos and Jurya Glansbeek for help sending the questionnaires and Kate McIntyre for language editing.

## Author Contributions

**Conceptualization:** Laura A. Bolte, Marjolein A. Y. Klaassen, Gerbrig J. Versteegen, Rinse K. Weersma, Floris Imhann.

**Data curation:** Laura A. Bolte, Marjolein A. Y. Klaassen, Valerie Collij, Arnau Vich Vila, Jingyuan Fu, Taco A. van der Meulen, Jacco J. de Haan, Aafje Dotinga, Alexandra Zhernakova, Cisca Wijmenga, Rinse K. Weersma, Floris Imhann.

**Formal analysis:** Laura A. Bolte, Marjolein A. Y. Klaassen.

**Funding acquisition:** Jingyuan Fu, Alexandra Zhernakova, Cisca Wijmenga, Rinse K. Weersma.

**Investigation:** Laura A. Bolte, Marjolein A. Y. Klaassen, Gerbrig J. Versteegen, Aafje Dotinga, Floris Imhann.

**Methodology:** Laura A. Bolte, Marjolein A. Y. Klaassen, Gerbrig J. Versteegen, Rinse K. Weersma, Floris Imhann.

**Project administration:** Valerie Collij.

**Resources:** Valerie Collij, Taco A. van der Meulen, Jacco J. de Haan, Alexandra Zhernakova, Cisca Wijmenga, Rinse K. Weersma.

**Software:** Valerie Collij.

**Supervision:** Arnau Vich Vila, Rinse K. Weersma, Floris Imhann.

**Visualization:** Laura A. Bolte, Floris Imhann.

**Writing – original draft:** Laura A. Bolte, Marjolein A. Y. Klaassen, Valerie Collij, Floris Imhann.

**Writing – review & editing:** Arnau Vich Vila, Jingyuan Fu, Taco A. van der Meulen, Jacco J. de Haan, Gerbrig J. Versteegen, Aafje Dotinga, Alexandra Zhernakova, Cisca Wijmenga, Rinse K. Weersma.

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
