## [Decision Letter · Decision Letter 0]

4 Jan 2021

PONE-D-20-24145

Patient attitudes towards faecal sampling for gut microbiome studies and clinical care reveal positive engagement and room for improvement

PLOS ONE

Dear Dr. Bolte,

Thank you for submitting your manuscript to PLOS ONE. After careful consideration, we feel that it has merit but does not fully meet PLOS ONE’s publication criteria as it currently stands. Therefore, we invite you to submit a revised version of the manuscript that addresses the points raised during the review process.

We look forward to receiving your revised manuscript.

Kind regards,

Nikhil Pai, BSc, MD

Academic Editor

PLOS ONE

Additional Editor Comments (if provided):

Thank you for your clear, and practical manuscript assessing the attitudes of patients towards faecal sampling for gut microbiome studies. The results of this paper are derived from an impressive survey sample, and the practical guidance offered in the conclusion of this piece are valuable. Please attend to the minor revisions suggested by the Reviewer. I also want to personally thank you for your patience. It has been challenging to find reviewers for certain publications, and this is not reflective of the quality of the work that you have submitted. Kindly attend to these minor revisions and I look forward to seeing a revised submission for assessment. Again, our thanks for your patience with this process.

Journal Requirements:

2) Please state whether the questionnaire was validated before data was collected.

3) We note that you have indicated that data from this study are available upon request. PLOS only allows data to be available upon request if there are legal or ethical restrictions on sharing data publicly. For information on unacceptable data access restrictions, please see http://journals.plos.org/plosone/s/data-availability#loc-unacceptable-data-access-restrictions.

4) PLOS requires an ORCID iD for the corresponding author in Editorial Manager on papers submitted after December 6th, 2016. Please ensure that you have an ORCID iD and that it is validated in Editorial Manager. To do this, go to ‘Update my Information’ (in the upper left-hand corner of the main menu), and click on the Fetch/Validate link next to the ORCID field. This will take you to the ORCID site and allow you to create a new iD or authenticate a pre-existing iD in Editorial Manager. Please see the following video for instructions on linking an ORCID iD to your Editorial Manager account: https://www.youtube.com/watch?v=_xcclfuvtxQ

5) Please include captions for your Supporting Information files at the end of your manuscript, and update any in-text citations to match accordingly. Please see our Supporting Information guidelines for more information: http://journals.plos.org/plosone/s/supporting-information.

6) Thank you for stating the following in the Competing Interests section:

[Floris Imhann received a speaker fee from Abbvie. Rinse Weersma received speaker

fees from Abbvie, MSD, and Boston Scientific, a consulting fee from Takeda

Pharmaceuticals and unrestricted research grants from Pfizer, Takeda, Ferring and

Tramedico.].

Reviewers' comments:

Reviewer's Responses to Questions

**Comments to the Author**

1. Is the manuscript technically sound, and do the data support the conclusions?

Reviewer #1: Yes

Reviewer #2: Yes

2. Has the statistical analysis been performed appropriately and rigorously? 

Reviewer #1: I Don't Know

Reviewer #2: Yes

3. Have the authors made all data underlying the findings in their manuscript fully available?

Reviewer #1: Yes

Reviewer #2: Yes

4. Is the manuscript presented in an intelligible fashion and written in standard English?

Reviewer #1: Yes

Reviewer #2: Yes

5. Review Comments to the Author

Reviewer #1: Dear authors

Thank you for writing such a clear and interesting article on the patient perspective towards stool collection. I have a few minor comments

1. introduction

Paragraph 1: Authors state ‘little is known about participant perspectives on collecting

faecal samples for microbiome research and future care, with available literature

currently limited to several studies examining participant experiences with the faecal

occult blood test (FOBT) used in colorectal cancer screening, the results of which mainly

capture experiences coloured by the fear of having cancer’ and reference 2 other studies.

I agree that literature on this topic is limited however it is not only limited to FOBT studies. The Lecky et al study for example examined patients both with and without previous experience of providing a stool sample and does not state that recruited participants had collected samples for FOBT.

Other work in this arena, not just linked to FOBT, also include

McNulty, C.A., Lasseter, G., Newby, K. et al. Stool submission by general practitioners in SW England - when, why and how? A qualitative study. BMC Fam Pract 13, 77 (2012). https://doi.org/10.1186/1471-2296-13-77

Breanna McSweeney, Jessica R. Allegretti, Monika Fischer, Huiping Xu, Karen J. Goodman, Tanya Monaghan, Carmen McLeod, Benjamin H. Mullish, Elaine O. Petrof, Emmalee L. Phelps, Roxana Chis, Abby Edmison, Angela Juby, Ralph Ennis-Davis, Brandi Roach, Karen Wong & Dina Kao (2020) In search of stool donors: a multicenter study of prior knowledge, perceptions, motivators, and deterrents among potential donors for fecal microbiota transplantation, Gut Microbes, 11:1, 51-62, DOI: 10.1080/19490976.2019.1611153

Please make this more clear in your introduction.

2. Discussion paragraphs 2 and 3

These paragraphs are more suited to introducing the topic and the need for stool collection rather than discussing the findings of this study and its relevance and/or implications

3. As previously mentioned, other research has been carried out in this areas therefore the discussion would benefit from more comparison to this literature highlighting similarities and differences and potential reasons for this.

Reviewer #2: The topic of the manuscript is interesting and important to tell researchers who are going to collect stool samples. This study highlighted the difficulty for achieving stool sample for microbial study. The “dirty” is one of the important factors affecting the attitudes to response the survey and collect stool sample. The lower response rate for this survey is the main limitation of this study, which might be the bias of the survey. However, authors did not clearly state this difficulty in INTRODUCTION or discuss this pint in DISCUSSION. As I know, the similar survey on patients’ or physicians’ attitude to process fecal transplant have the similar difficulty in practice. These should be discussed for supporting authors’ findings. The following reports should be included for discussion:

Patients' views on fecal microbiota transplantation: an acceptable therapeutic option in inflammatory bowel disease? Eur J Gastroenterol Hepatol 2017.

Perceptions of fecal microbiota transplantation for Clostridium difficile infection: factors that predict acceptance. Ann Gastroenterol. 2017.

The recognition and attitudes of postgraduate medical students toward fecal microbiota transplantation: a questionnaire study. Therap Adv Gastroenterol. 2019

Minor comments as below:

1. In introduction, microbiota is target for fecal microbiota transplantation too. This is very important. Please add it in the sentence.

2. Methods - There were nearly 40 questions in your questionnaire. Why do you only calculate the five questions which you mentioned?

3. In table 1, the number of patients in “Maximum time patients want to store fecal samples in their freezer” should subtract 73 as shown in “Unpleasant to store fecal samples in home freezer?”. Seventy-three patients are unpleasant to store fecal samples in home freezer. So they are not necessary to consider the time to store fecal sample.

4. Results - 97.4% had collected a faecal sample for prior gut microbiome research projects. Is this data for question eight in the Supplementary Table 1? An error?

5. Results - According to your definition in the Supplementary Method, the patients with gastrointestinal disease includes “IBD willing + IBD unwilling + no identification number” participants. You mentioned in the results that 250 patients had GI-disorder. This may be a statistical error. In additon, this is a little different from the data in the questionnaire. Because you asked participants in question eight if they had been diagnosed with intestinal disease, and only 42 patients answered no.

6. Limitation should better not be included in conclusion. Please put it to the part of discussion.

7. Conclusions include too much content. Please try to use summative words in this part and put the current content to suitable part.

8. The structure of the manuscript is unconventional. Please refer the regular style of the journal.

6. PLOS authors have the option to publish the peer review history of their article (what does this mean?). If published, this will include your full peer review and any attached files.

Reviewer #1: No

Reviewer #2: No

---

## [Author Response · Author response to Decision Letter 0]

19 Feb 2021

Additional journal requirements

1) PLOS ONE's style requirements: The journal’s requirements were adopted.

2) Questionnaire validation: The questionnaire was specifically designed for this study together with the Department of Health Psychology of the University Medical Center Groningen. The questionnaire has not been validated in a separate study.

3) Data availability and reproducibility: The summary statistics for each question in the questionnaire are available in Supplementary Tables S1 and S2. Using these statistics, all tests in this study can be replicated. Our cohorts comprise specific groups of patients from a specific geographical area and the content of the questionnaire comprises household composition, education level and home details about the toilet and the freezer. As a consequence, individual level data could lead to identification of patients. Therefore, European privacy law prohibits us from making individual level data publicly available. For collaborations please contact the Research Office Lifelines (research@lifelines.nl) and the Medical Ethics Review Board of the UMCG (metc@umcg.nl). 

4) ORCID iD of corresponding author: In the editorial manager we were only able to add the ORCID iD of the submitting author (Laura Bolte). The ORCID iD of the corresponding author (Floris Imhann) is: 0000-0001-5278-903X. 

5) Supporting file captions and citations: Supporting file captions and file names have been adapted according to the journals’ requirements and matched with in-text citations. 

6) Competing Interests: Floris Imhann received a speaker fee from Abbvie. Rinse Weersma received speaker fees from Abbvie, MSD, and Boston Scientific, a consulting fee from Takeda Pharmaceuticals and unrestricted research grants from Pfizer, Takeda, Ferring and Tramedico. This does not alter our adherence to PLOS ONE policies on sharing data and materials.

Reviewer 1

Q.1.1: Thank you for writing such a clear and interesting article on the patient perspective towards stool collection. I have a few minor comments. 

Introduction, Paragraph 1: Authors state ‘little is known about participant perspectives on collecting faecal samples for microbiome research and future care, with available literature currently limited to several studies examining participant experiences with the faecal occult blood test (FOBT) used in colorectal cancer screening, the results of which mainly capture experiences coloured by the fear of having cancer’ and reference 2 other studies. I agree that literature on this topic is limited however it is not only limited to FOBT studies. The Lecky et al study for example examined patients both with and without previous experience of providing a stool sample and does not state that recruited participants had collected samples for FOBT. Other work in this arena, not just linked to FOBT, also include: 

McNulty, C.A., Lasseter, G., Newby, K. et al. Stool submission by general practitioners in SW England - when, why and how? A qualitative study. BMC Fam Pract 2012, 13:77.

McSweeney B, Allegretti JR, Fischer, M et al. In search of stool donors: a multicenter study of prior knowledge, perceptions, motivators, and deterrents among potential donors for fecal microbiota transplantation, Gut Microbes 2020, 11:1, 51-62.

Please make this more clear in your introduction.

R.1.1: We thank the reviewer for the positive feedback and suggestions to improve our manuscript. We agree that our findings connect well with other disciplines such as the diagnosis of intestinal infectious diseases as part of clinical routine, or the search for stool donors for fecal microbiota transplantation. We now refer to these articles in the introduction. 

Page 3, lines 9-12: 

“Moreover, as the clinical interest grows in the use of fecal microbiota transplantation (FMT) for dysbiosis-related disorders such as recurrent Clostridium difficile associated diarrhea and IBD, so will the need for voluntary stool donors.[10–12]”

[10] McSweeney B, Allegretti JR, Fischer M, Xu H, Goodman KJ, Monaghan T, et al. In search of

 stool donors: a multicenter study of prior knowledge, perceptions, motivators, and deterrents 

 among potential donors for fecal microbiota transplantation. Gut Microbes 2019; 11: 51-62. 

[11] Moayyedi P, Surette MG, Kim PT, Libertucci J, Wolfe M, Onischi C, et al. Fecal Microbiota 

 Transplantation Induces Remission in Patients With Active Ulcerative Colitis in a Randomized 

 Controlled Trial. Gastroenterology 2015; 149: 102-109.e6. 

[12] Vrieze A, Nood EV, Holleman F, Salojärvi J, Kootte RS, Bartelsman JFWM, et al. Transfer of 

 Intestinal Microbiota From Lean Donors Increases Insulin Sensitivity in Individuals With 

 Metabolic Syndrome. Gastroenterology 2012; 143: 913-916.e7. 

Page 3, lines 18-24: 

“Other studies report patient barriers to fecal sample collection in general practice, including difficulty with the process, embarrassment and concerns around hygiene. [14, 15] Despite these barriers, most patients do provide fecal samples in the clinical setting, because they are unwell and have been recommended to do so. [15] While personal gain has been identified as the main motive for collecting and returning a stool sample in clinical care,[14] there is no direct personal benefit for voluntary stool donors for FMT or microbiome research, who may face similar barriers.”

[14] Lecky DM, Hawking MK, McNulty CA. Patients’ perspectives on providing a stool sample to

 their GP: a qualitative study. Br J Gen Pract 2014; 64: e684–e693. 

[15] McNulty CA, Lasseter G, Newby K, Joshi P, Yoxall H, Kumaran K, et al. Stool submission by

 general practitioners in SW England - when, why and how? A qualitative study. BMC Family 

 Practice 2012; 13: 77. 

Q.1.2: Discussion paragraphs 2 and 3: These paragraphs are more suited to introducing the topic and the need for stool collection rather than discussing the findings of this study and its relevance and/or implications.

R.1.2: We agree with the reviewer that the need for stool collection is more suitable to introduce the topic and have moved these paragraphs to the introduction. 

Page 3, lines 5-12: 

“The results of these studies hold great promise for clinical applications that include the use of microbiome features as diagnostic biomarkers,[5] determinants of disease activity,[3] and predictors of individual drug response.[6,7] The microbiome itself may also be a treatment target for prebiotic, probiotic, antibiotic and dietary interventions. [2,8,9] Moreover, as clinical interest grows in the use of fecal microbiota transplantation (FMT) for dysbiosis-related disorders such as recurrent Clostridium difficile infection and IBD, so will the need for voluntary donors.[10–12]” 

Q.1.3: As previously mentioned, other research has been carried out in these areas therefore the discussion would benefit from more comparison to this literature highlighting similarities and differences and potential reasons for this.

R.1.3: We agree with the reviewer and added a comparison of the identified attitudes towards fecal sampling for gut microbiome research in our study with previous reports on the perspectives on collecting or donating fecal samples for clinical care or FMT.

Page 12, lines 16-19:

“This is in line with a previous report of the attitudes of 400 patients towards participation in clinical trials conducted at an internal medicine ward.[25] Positive feedback on how FMT can help patients were also motivators to donate fecal samples for FMT.[10]”

Page 13, line 7-13:

“McSweeney et al. also identified altruism as the main motive to donate fecal samples for FMT and many patients who were willing to donate fecal samples said to do so to help those who were ill and contribute to progress in scientific research. [10] As expected, this differs from the motives of patients who collect or receive fecal samples as part of their clinical care, where personal gain is the main incentive. [14] Despite concerns around hygiene, logistics and privacy, most patients return their faecal sample to their doctor because they were recommended to do so for their own health.[15]”

Page 15, line 5-12:

“Other studies identified barriers towards fecal sampling in clinical care, including difficulty with the collection process, lack of information given by doctors and inability to return the sample to the institution. [14, 15] The difficulty of collecting the fecal sample was one of the major factors impacting FOBT response in a South African study. [29] Higher donation frequency, logistics of collection or transport of fecal samples, screening process, lack of public awareness and negative social perception were identified as deterrents to donate stool for FMT. [10]” 

Page 15, line 20 – Page 16, line 2:

“Similar to our findings, studies indicate that patients collecting fecal samples for clinical care value an information leaflet provided with the stool collection kit [14] and that screening compliance for the FOBT is significantly improved when patients obtain this information. [30]”

[10] McSweeney B, Allegretti JR, Fischer M, Xu H, Goodman KJ, Monaghan T, et al. In search of 

 stool donors: a multicenter study of prior knowledge, perceptions, motivators, and deterrents 

 among potential donors for fecal microbiota transplantation. Gut Microbes 2019; 11:51-62.

[14] Lecky DM, Hawking MK, McNulty CA. Patients’ perspectives on providing a stool sample to 

 their GP: a qualitative study. Br J Gen Pract 2014; 64: e684–e693. 

[15] McNulty CA, Lasseter G, Newby K, Joshi P, Yoxall H, Kumaran K, et al. Stool submission by

 general practitioners in SW England - when, why and how? A qualitative study. BMC Family 

 Practice 2012; 13: 77. 

[29] Price CL, Szczepura AK, Gumber AK, Patnik J. Comparison of breast and bowel cancer

 screening uptake patterns in a common cohort of South Asian women in England. BMC Health

 Services Research 2010; 10: 103. 

[30] O’Carroll RE, Steele RJ, Libby G, Brownlee L, Chambers JA. Anticipated regret to increase 

 uptake of colorectal cancer screening in Scotland (ARTICS): study protocol for a randomised 

 controlled trial. BMC Public Health 2013; 13: 849. 

Reviewer 2

Q.2.1: The topic of the manuscript is interesting and important to tell researchers who are going to collect stool samples. This study highlighted the difficulty for achieving stool sample for microbial study. The “dirty” is one of the important factors affecting the attitudes to response the survey and collect stool sample. The lower response rate for this survey is the main limitation of this study, which might be the bias of the survey. However, authors did not clearly state this difficulty in INTRODUCTION or discuss this point in DISCUSSION. As I know, the similar survey on patients’ or physicians’ attitude to process fecal transplant have the similar difficulty in practice. These should be discussed for supporting authors’ findings. The following reports should be included for discussion:

Patients' views on fecal microbiota transplantation: an acceptable therapeutic option in inflammatory bowel disease? Eur J Gastroenterol Hepatol 2017.

Perceptions of fecal microbiota transplantation for Clostridium difficile infection: factors that predict acceptance. Ann Gastroenterol. 2017.

The recognition and attitudes of postgraduate medical students toward fecal microbiota transplantation: a questionnaire study. Therap Adv Gastroenterol. 2019

R.2.1: We thank the reviewer for the thorough inspection of our manuscript. We agree that the non-response bias should be stated more clearly as a limitation in the discussion section. Moreover, we agree that the above-mentioned literature should be described and compared to our results more elaborately in the discussion. 

Therefore, we have largely rewritten our discussion including the literature and limitations mentioned by the reviewer. 

Page 11, line 15 – Page 12, line 11: 

“Other studies assessed patient willingness to receive or donate stool samples for FMT, the transfer of fecal material containing microbiota from a healthy donor into a diseases patient. One study found that 77% of patients visiting the gastroenterologist would undergo FMT if medically indicated. [22] Conversely, only 36.9% of IBD patients were willing to undergo FMT in a report by Zeitz et al. [23] Familiarity with the gut microbiome might contribute to a higher willingness to participate in gut microbiome research in our study compared to FMT. Only 46.5% of IBD patients [23] and 12% of patients visiting a gastroenterologist [22] knew about FMT. Interestingly, the willingness rate of IBD patients to undergo FMT almost doubled after an information leaflet was provided. [23] Recognition of FMT in postgraduate medical students was equally low. [24] While nearly half of the students had not heard about FMT, the majority recognized that disrupting and restoring gut microbiota played an important role in the pathogenesis and prevention of diseases. In the same study, willingness to undergo FMT or donate samples was significantly higher among those who were familiar with FMT. [24]”

[22] Park et al. Perceptions of fecal microbiota transplantation for Clostridium difficile infection: factors that predict acceptance. Ann Gastroenterol 2017; 30: 83–88. 

[23] Zeitz et al. Patients’ views on fecal microbiota transplantation: an acceptable therapeutic option in inflammatory bowel disease? Eur J Gastroenterol Hepatol 2017; 29: 322–330.

[24] Wu et al. The recognition and attitudes of postgraduate medical students toward fecal 

 microbiota transplantation: a questionnaire study. Therap Adv Gastroenterol 2019; 12: 

 1756284819869144. 

Page 13, lines 14-20:

“Similarly, studies showed that the majority of patients would undergo FMT if it was medically indicated and recommended by their doctor [22] and willingness to undergo FMT was positively associated with disease severity and previous TNF-treatment in IBD patients. [23, 28] Naturally, recommendation by your own doctor is not part of when recruiting volunteers for microbiome research or healthy donors for FMT, making it even more crucial to inform volunteers about the process and remove any barriers in order to obtain sufficient sample sizes or to guarantee cost-effectiveness.”

[28] Kahn et al. Fecal bacteriotherapy for ulcerative colitis: patients are ready, are we? Inflamm Bowel Dis 2012; 18: 676–684. 

Page 14, lines 11-19:

“Fear of infectious diseases and disgust about the procedure were also identified as the most common concerns of patients to undergo FMT. [22, 23] This is underlined by the finding that IBD patients would choose a colonoscopy as the preferred route of FMT rather than an enema or nasogastric tube. [22, 23] Even post-graduate medical students considered donating feces troublesome because it hampered their privacy and also showed concerns about the acceptability among patients. [24] Privacy was not a big concern in our study. While screening for FMT requires the donor to provide a lot of private information as not all stool samples are suitable, participants in gut microbiome research might feel more anonymous.”

Page 15, lines 20-22:

“This may be particularly important in populations with a lower health illiteracy, with a study showing that higher education levels are strong predictors of FMT acceptance in patients. [22]”

Page 16, lines 7-11: 

“Another study even showed that having children and being married were strong predictors of FMT acceptance in patients. [22] It is possible that patients with children are more likely to embrace FMT, even though it may be unappealing in nature, because of their responsibility towards their family. [27]”

[27] Prainsack and Buyx. A solidarity-based approach to the governance of research 

biobanks. Medical Law Review 2013; 21: 71–91.

Page 16, line 12 – Page 17, line 5: 

“Strengths and limitations

Our questionnaire study was limited by knowing only the answers of the respondents. IBD patients who previously disagreed to participate in our gut microbiome studies (IBD-Unwilling, n=83) were also less likely to respond to the questionnaire (n=3), making it difficult to assess their reasons to refuse participation. Overall, the 39.1% response rate to our questionnaire is in line with the recognised average response rate for postal surveys (40%). [31] Another survey of IBD patients to investigate their perspectives on FMT, obtained a response rate of 31.4%. [23] The positive attitudes towards fecal sample collection in our study may not always be representative of other patients, and attitudes may differ depending on the reason for stool sample collection, e.g. samples collected for research vs. those collected for diagnosis of a potential disease (a process that may be accompanied by fear), or the health care setting, e.g. secondary vs. routine primary care. The strength of our study is that we were able to obtain information on the attitudes to, motives for and barriers to participation in gut microbiome research of 780 patients with different disorders and healthy volunteers, which has not been assessed to date. We obtained enough information to formulate the following conclusions and recommendations for both gut microbiome researchers and clinicians.”

Q.2.2: Minor comments as below: In introduction, microbiota is a target for fecal microbiota transplantation too. This is very important. Please add it in the sentence.

R.2.2: We agree with the reviewer. The other reviewer has also addressed this issue (Q.1.1). We have now added the prospect of fecal transplantation in the introduction. 

Page 3, lines 9-12: 

“Moreover, as clinical interest grows in the use of fecal microbiota transplantation (FMT) for dysbiosis-related disorders such as recurrent Clostridium difficile infection (CDI) and IBD, so will the need for voluntary donors.”

Q.2.3: Methods - There were nearly 40 questions in your questionnaire. Why do you only calculate the five questions which you mentioned?

R.2.3: For all factors assessed by the questionnaire (38 questions), we calculated the average (SD) and counts (%), respectively (S1 Table). This way, an overview of the participants’ answers on all the assessed factors was displayed, including views on the participants’ home situation, whether they felt free to refuse participation, their intrinsic motivation to collect feces, and logistic aspects such as type of toilet, freezer and time they had to wait until the sample was picked up. 

After presenting these descriptive statistics of the participants’ perspectives on the fecal sample collection, we wanted to answer a limited number of specific questions that we think would help microbiome researchers conduct better studies, rather than perform statistical analyses comparing every question to every factor in order to find any statistically significant differences.

Q.2.4: In table 1, the number of patients in “Maximum time patients want to store fecal samples in their freezer” should subtract 73 as shown in “Unpleasant to store fecal samples in home freezer?”. Seventy-three patients are unpleasant to store fecal samples in home freezer. So they are not necessary to consider the time to store fecal sample.

R.2.4: We understand the remark of the reviewer and have subtracted these 73 participants from this equation in the results, Table 1 (page 10) and Supplementary Table 1. 

Page 8, line 9 – 11: 

“However, while most patients were willing to store a stool sample in their freezer, many were only willing to do so for a brief period of time: maximum 1 to 3 days (14.6%), 1 week (25.3%), or 2 to 4 weeks (11.1%). 40.9% said that they did not mind storing faecal samples for a longer time.”

Q.2.5: Results - 97.4% had collected a faecal sample for prior gut microbiome research projects. Is this data for question eight in the Supplementary Table 1? An error?

R.2.5: Indeed, 97.4% had collected a faecal sample for gut microbiome research before (summing up answers A to D in question 8 “Did you ever participate in research for which you had to collect and deep-freeze a small stool sample?”: A) yes, 1 time + B) yes, 2-5 times + C) yes, 6-10 times + D) yes, >10 times). Only in the IBD cohort we were also able to send questionnaires to patients who previously refused to participate in gut microbiome studies, of which 3 patients responded (IBD-Unwilling, n=3). The other cohorts (melanoma, Sjögren’s syndrome, SLE) were comprised of gut microbiome research participants. We agree that this needs a better explanation in the methods section. 

Page 4, lines 17-30: 

A questionnaire (S1 Table) was sent in January 2017 to 772 patients who had previously been recruited at the University Medical Center Groningen in the Netherland for gut microbiome studies for which they needed to provide a faecal sample. These patients had been included in four disease-specific cohorts: IBD (n=660), melanoma (n=9), Sjögren’s syndrome (n=55) and systemic lupus erythematosus (SLE) (n=48) (Fig 1). The latter three cohorts only comprised participants who joined the gut microbiome studies. The questionnaire was aimed at obtaining their experiences and identifying barriers encountered during the collection process. With the IBD cohort, we were also able to send out questionnaires to patients who previously refused to participate in gut microbiome research. Questionnaire recipients in the IBD cohort therefore comprised both patients previously willing to collect a stool sample for research (n=577, IBD-Willing) and patients previously not willing to do so (n=83, IBD-Unwilling), indicating a willingness rate of 87.4% of the IBD microbiome study prior to this survey.”

We also asked if patients were willing to participate again after their experiences. 

Page 11, lines 10-14: 

“Our study has demonstrated that stool sample collection for gut microbiome studies and future clinical applications is acceptable to the majority of IBD patients and even to population controls. Most IBD patients (87.4%) were willing to participate in our previous stool sample collection (IBD-Willing, n = 577) and most respondents (82.9%) and interviewees (95.6%) indicated that they were willing to participate again.”

Q.2.6: Results - According to your definition in the Supplementary Method, the patients with gastrointestinal disease includes “IBD willing + IBD unwilling + no identification number” participants. You mentioned in the results that 250 patients had GI-disorder. This may be a statistical error. In addition, this is a little different from the data in the questionnaire. Because you asked participants in question eight if they had been diagnosed with intestinal disease, and only 42 patients answered no.

R.2.6: We thank the reviewer for his remark. This calculation was indeed performed based on the participants belonging to the IBD cohort. The reviewer is referring to the answers to question 6 in the questionnaire (S1 Table), where we asked the participants specifically about the presence of any intestinal diseases. Indeed, this data is different from the definition by cohort (IBD cohort vs. No IBD cohort). We have now also performed the analysis using question 6 to define presence or absence of intestinal disease: There was no correlation between GI-disease (Q6) and willing to participate in future faecal sample collection (p = 0.564, Table 1, page 9). 

Q2.7: Limitation should better not be included in conclusion. Please put it to the discussion.

R2.7: The limitations now appear at the end of the discussion on page 16, lines 1-13.

Q2.8: Conclusions include too much content. Please try to use summative words in this part and put the current content to suitable part.

R.2.8: The conclusion has been adapted. Page 25, lines 7-21: 

“Targeting the gut microbiome will soon be part of the diagnostic process and treatment of IBD and other diseases that are associated with microbial dysbiosis [5,6,31,32] requiring repeated sampling from patients.[33] Here, we assessed the perspectives of patients and healthy volunteers on faecal sampling for gut microbiome research. We derive the following recommendations for gut microbiome researchers and clinicians: 

(1) Gut microbiome researchers setting up new cohorts or clinicians trying new faecal tests should not shy away from doing so.

(2) Gut microbiome researchers and clinicians should explain why their collection protocol was designed in a specific way. 

(3) In studies where a time-series of many stool samples needs to be collected, researchers should consider providing participants with a small freezer. 

(4) Researchers and clinicians should inform participating patients and healthy volunteers about the outcome of the research.” 

Q.2.9: The structure of the manuscript is unconventional. Please refer the regular journal style.

R.2.8: We have changed the structure of our manuscript accordingly.

---

## [Decision Letter · Decision Letter 1]

18 Mar 2021

Patient attitudes towards faecal sampling for gut microbiome studies and clinical care reveal positive engagement and room for improvement

PONE-D-20-24145R1

Dear Dr. Bolte,

We’re pleased to inform you that your manuscript has been judged scientifically suitable for publication and will be formally accepted for publication once it meets all outstanding technical requirements.

Kind regards,

Nikhil Pai, BSc, MD

Academic Editor

PLOS ONE

Additional Editor Comments (optional):

Reviewers' comments:

Reviewer's Responses to Questions

**Comments to the Author**

1. If the authors have adequately addressed your comments raised in a previous round of review and you feel that this manuscript is now acceptable for publication, you may indicate that here to bypass the “Comments to the Author” section, enter your conflict of interest statement in the “Confidential to Editor” section, and submit your "Accept" recommendation.

Reviewer #2: All comments have been addressed

2. Is the manuscript technically sound, and do the data support the conclusions?

Reviewer #2: Yes

3. Has the statistical analysis been performed appropriately and rigorously? 

Reviewer #2: Yes

4. Have the authors made all data underlying the findings in their manuscript fully available?

Reviewer #2: Yes

5. Is the manuscript presented in an intelligible fashion and written in standard English?

Reviewer #2: Yes

6. Review Comments to the Author

Reviewer #2: The revision is nice. All comments have been addressed.

This study highlighted the difficulty for achieving stool sample for microbial study. The “dirty” is one of the important factors affecting the attitudes to response the survey and collect stool sample. This kind of study focus on stool should be put on the better position for more attention.

7. PLOS authors have the option to publish the peer review history of their article (what does this mean?). If published, this will include your full peer review and any attached files.

Reviewer #2: No

---

## [Editor Report · Acceptance letter]

31 Mar 2021

PONE-D-20-24145R1 

Patient attitudes towards faecal sampling for gut microbiome studies and clinical care reveal positive engagement and room for improvement 

Dear Dr. Bolte:

I'm pleased to inform you that your manuscript has been deemed suitable for publication in PLOS ONE. Congratulations! Your manuscript is now with our production department. 

Kind regards, 

on behalf of

Dr. Nikhil Pai 

Academic Editor

PLOS ONE